# The Position of the Virtual Hinge Axis in Relation to the Maxilla in Digital Orthognathic Surgery Planning—A k-Means Cluster Analysis

**DOI:** 10.3390/jcm12103582

**Published:** 2023-05-21

**Authors:** Thomas Stamm, Moritz Kanemeier, Dieter Dirksen, Claudius Middelberg, Ariane Hohoff, Johannes Kleinheinz, Jonas Q. Schmid

**Affiliations:** 1Department of Orthodontics, University of Münster, 48149 Münster, Germany; 2Department of Prosthetic Dentistry and Biomaterials, University of Münster, 48149 Münster, Germany; 3Department of Cranio-Maxillofacial Surgery, University of Münster, 48149 Münster, Germany

**Keywords:** orthognathic surgery, virtual articulator, mounting, hinge axis, digital planning

## Abstract

The aim of this study was to investigate a possible relation between skeletal phenotypes and virtual mounting data in orthognathic surgery patients. A retrospective cohort study including 323 female (26.1 ± 8.7 years) and 191 male (27.9 ± 8.3 years) orthognathic surgery patients was conducted. A k-means cluster analysis was performed on the mounting parameters: the angle α between the upper occlusal plane (uOP) and the axis orbital plane (AOP); the perpendicular distance (AxV) from the uOP to the hinge axis; and the horizontal length (AxH) of the uOP from upper incisor edge to AxV, with subsequent statistical analysis of related cepalometric values. Three clusters of mounting data were identified, representing three skeletal phenotypes: (1) balanced face with marginal skeletal class II or III and $\alpha =8^{\circ }$, AxV = 36 mm and AxH = 99 mm; (2) vertical face with skeletal class II and $\alpha =11^{\circ }$, AxV = 27 mm and AxH = 88 mm; (3) horizontal face with class III and $\alpha =2^{\circ }$, AxV = 36 mm and AxH = 86 mm. The obtained data on the position of the hinge axis can be applied to any digital planning in orthognathic surgery using CBCT or a virtual articulator, provided that the case can be clearly assigned to one of the calculated clusters.

## 1. Introduction

In the pre-digital era, orthognathic surgery was generally planned using plaster models. These were placed in mean value or semi-adjustable articulators and allowed a more or less individual position of the jaws in relation to the hinge axis, with equal pro-, latero- and opening movements. The mounting of the models can be carried out either by using a facebow or by using the averages of the Bonwill triangle and Balkwill angle. Bonwill described an equilateral triangle of 101.6 mm connecting the midpoints of the condyles and the contact point of the lower central incisors, based on the measurements of 4000 skulls and 6000 living individuals [1,2,3,4,5]. Balkwill was the first person to measure the forward and downward movement of the condyles during mouth opening and described an angle between the occlusal plane of the lower jaw and the plane of the Bonwill triangle [2,5]. Both procedures, i.e., the use of a facebow and the use of the Bonwill and Balkwill averages, are called arbitrary mounting. The use of a facebow is considered significantly more reliable than mounting the models with the Bonwill and Balkwill averages [6] and is therefore the standard method for analogue planning of orthognathic surgery using a mechanical articulator.

With the continuous evolution of CAD/CAM technologies in dentistry, different ideas have been introduced to virtually create a patient-specific situation of the facial and masticatory apparatus. Two main procedures are described: (a) the direct workflow with a virtual facebow, and (b) an indirect workflow with a desktop laboratory scanner that transfers a mechanical articulator to a virtual articulator [7]. The techniques used to create a virtual articulator include the use of arbitrary values [7], reconstruction of 2-dimensional data [8], stereophotogrammetry [9] and cone-beam computed tomography (CBCT) [10]. A comprehensive overview of mounting methods is provided by Lepidi and coworkers [7].

Although planning in orthognathic surgery has conventionally always been performed in the articulator, new methods do not use the virtual articulator approach, but focus on simulating osteotomies using CBCT [11]. The majority of these methods require the support of commercial vendors [12]. Furthermore, the hinge axis, if localized at all in the CBCT, plays only a minor role.

The simulation of rotational movements around the hinge axis is essential during orthognathic surgery planning. For this reason, we have developed a three-dimensional planning system for orthognathic surgery based on the principles of a virtual articulator: the Digital Münster Model Surgery (DMMS) system. The methodology of the DMMS has been published elsewhere [13], but for the understanding of the present work, the determination of the virtual hinge axis in relation to the maxilla using the DMMS is briefly repeated below.

The DMMS was developed based on conventional plaster surgery described by Ehmer et al. [14,15,16]. Similarly to conventional planning using the SAM 2P semi-adjustable articulator (SAM Präzisionstechnik GmbH, Gauting, Germany), the starting position is the virtual mounting, i.e., the correct alignment of the maxillary and mandibular scans relative to a reference plane and the hinge axis. For this purpose, the position of the physical SAM facebow (reference plane) was marked with metal markers on the patient’s skin as a standard procedure before a lateral cephalogram was taken (Figure 1, left).

A retrospective analysis of the marked radiographs showed that the position of the maxillary dentition in relation to the axis orbital plane (AOP) can be calculated using four landmarks: deepest point on the lower border of the meatus acusticus externus (Porion inferior; Pi), nasal support of the face-bow (Ns), upper incisor edge (Ie), and distobuccal cusp of the first permanent upper molar (Dc) [13]. Based on these landmarks, a virtual facebow can be constructed and projected onto the lateral cephalogram (Figure 1, right). The following values for mounting intraoral scans into a virtual space can then be obtained: 1. The angle α between the upper occlusal plane (uOP) and the AOP; 2. the vertical distance (AxV) between the arbitrary hinge axis (Ax) and its projection (Ax’) onto the occlusal plane; 3. the horizontal distance (AxH) between Ie and Ax’. These values are sufficient to align a maxillary scan in a virtual space in the same way as in the conventional SAM articulator with physical facebow and bite fork [11,13].

The software ImageJ [17] can be used to place the aforementioned landmarks (Figure 1) on a lateral cephalogram to construct Ax and calculate the mounting values α, AxV and AxH. In an effort to increase the efficiency of orthognathic surgery planning, the question arose of whether Bonwill and Balkwill averages are sufficient to define the position of the hinge axis in relation to the jaws. As Bonwill and Balkwill did not explicitly study patients who underwent orthognathic surgery, it was initially assumed that, due to the underrepresentation of severe skeletal malocclusions, the averages of Bonwill and Balkwill would be different from the mounting values found in orthognathic surgery patients. In addition, it was presumed that opposing skeletal malocclusions could cancel each other out in mean mounting values. To test this, the existing data of the digitally planned orthognathic surgeries were subjected to a cluster analysis. It was hypothesised that certain skeletal phenotypes, such as Angle class II, Angle class III, vertical face or horizontal face could be assigned to specific arbitrary mounting parameters. Such average mounting values would reduce planning time and could also be applied to CBCT data.

The aim of the study was, therefore, to investigate a possible relationship between preoperative cephalometric measurements and virtual mounting values in a large group of patients who underwent the digital orthognathic surgery workflow.

## 2. Materials and Methods

This retrospective cohort study received approval from the Ethics Commission of the Medical Faculty of the University of Münster, Germany (2021-120-f-S). The study took place at the Dental School of the University Hospital, Münster, Germany. All cases included in the study were planned with the DMMS in the Department of Orthodontics and underwent orthognathic surgery in the Department of Cranio-Maxillofacial Surgery between February 2018 and December 2022. A further inclusion criterion was the completeness of the preoperative planning documents, i.e., the mounting values (Figure 1), preoperative cephalometric analyses (Figure 2, Table 1) and planning protocols. Exclusion criteria were patients with orofacial syndromes, patients who needed comprehensive prosthetic rehabilitation and also distraction osteogenesis in general.

Due to the large number of medical records that needed to be screened, the initial step was to write Python scripts to automatically read out the relevant data and to check for completeness. In the next step, the data were aggregated on a patient-by-patient basis to perform a cluster analysis. Subsequently, the patients assigned to each cluster were evaluated with respect to their skeletal configuration to identify cluster-specific skeletal phenotypes. Finally, the mounting values were compared with the Bonwill and Balkwill averages, assuming that one arm of the Bonwill triangle can be compared with the distance Ie-Ax (Figure 1) and the Balkwill angle with the angle between the occlusal plane and the distance Ie-Ax.

### Statistical Analysis

A k-means cluster analysis was performed to group similar data points (clusters) of the mounting variables (α, AxV and AxH) together and to discover a possible underlying pattern. For this purpose, the open source software R (version 4.2.2, R Foundation for Statistical Computing, Vienna, Austria) together with the R package “factoextra” was employed. After scaling the variables, this algorithm subdivides the data into a pre-selected number of groups (i.e., clusters) with minimized intra-cluster distances. The optimal number of clusters was determined using a heuristic criterion, which here employs a curve where the total sum of intra-cluster squared distances is plotted against the number of clusters. The "elbow criterion" was applied; i.e., the number of clusters at which the curve starts to flatten was chosen.

Each identified cluster represented a subgroup of patients characterized by individual cephalometric values (Table 1). The cephalometric inter-cluster differences were examined using a chi-square test and analysis of variance (ANOVA) post hoc Tukey test. To assess cluster-specific differences in age and mounting data, the Mann–Whitney U test was used. The datasets were analyzed using IBM SPSS Statistics 29 (IBM Corp., Armonk, NY, USA).

## 3. Results

From 568 cases planned with the DMMS system in 2018–2022 and thus eligible for inclusion, 37 cases met the exclusion criteria and surgery was planned twice for 17 patients, so that 514 cases were finally included. The study group consisted of 323 female (mean age 26.1 ± 8.7 years) and 191 male (mean age 27.9 ± 8.3 years) patients. The male patients were significantly older than the female patients (Mann–Whitney U; p< 0.001).

With regard to the mounting data, it was found that the values differed significantly between female and male patients. For the horizontal (AxH) and vertical distances (AxV), significantly larger values were measured in males (p< 0.001), but the angle α was significantly smaller (p< 0.011) than in females (Table 2).

### 3.1. Cluster Identification

As a result of the k-means cluster analysis, different clusters were identified, in which the variables angle α, distance AxV, and AxH were grouped. The elbow method was used to determine the cut-off point, which indicated that the number of clusters selected should be three. The resulting clusters are visualized in Figure 3.

Table 3 shows the mounting data, age and gender of the study group according to the three clusters. There was a statistically significant difference in terms of gender distribution between the clusters (chi-square, p<0.001, Cramer’s V = 0.451). There was no statistically significant difference regarding age between clusters (ANOVA, p>0.05).

Cluster 2 represents the largest subgroup, with 204 patients, having a disproportionately large number of female patients. Here, the largest angulation of the upper occlusal plane to AOP, with the lowest values for AxV, are found. This pattern was initially attributed to a vertical face type with decreased posterior face height.

Cluster 3 is the second-largest subgroup and is predominantly characterized by the smallest α with the lowest value for AxH. This pattern was attributed to a horizontal face type with a shortened midface.

Cluster 1 summarizes a group of patients characterized only by a large value of AxH. It is also remarkable that more than twice as many male patients as female patients are represented in this group. This pattern was attributed to an average face type. All values between clusters were statistically significantly different (ANOVA, p<0.001), except AxV between clusters 1 and 3.

### 3.2. Cluster-Specific Skeletal Phenotypes

To assess cluster-specific skeletal phenotypes, an inter-cluster comparison of the cephalometric values was performed. The one-way ANOVA (post hoc Tukey) test revealed that only 9 of 22 cephalometric variables were statistically different between each cluster (Table 4). Other variables that were statistically significantly different between two clusters only were excluded.

The nine included variables are representative for the entire cephalometric analysis: three variables describe the mandible, two the maxilla, one the maxillo-mandibular relation, and three individual values characterize the facial type, the dental area and the soft tissue profile. It is noticeable that only one angle (facial axis) describes the vertical pattern. Other important vertical variables, such as lower facial height or mandibular plane, were not suitable to differentiate between all three clusters.

The most prominent cluster is cluster 2. In this cluster, predominantly female patients were included, and seven of nine variables indicate a vertical face combined with skeletal class II. There were also more female patients in cluster 3 but with variables indicating a horizontal face combined with skeletal class III (facial axis, facial depth, SNB). It appears that all patients who are not clearly skeletal class II or III are grouped in cluster 1. Seven of nine variables in cluster 1 are classified as balanced, one indicates class II, and one class III. The main difference in relation to the other clusters is that more male than female patients are represented here, and thus, larger values for AxH and AxV are present.

The distance Ie-Ax as an approximation of the Bonwill arm and the calculation of the Balkwill angle is given in Table 5. It could be shown that all values differ significantly in the inter-cluster comparison. Only cluster 1 is close to the Bonwill and Balkwill averages. However, this is not representative for all surgical patients, as this is the numerically smallest cluster. Despite the wide range (18–25°) of the Balkwill angle, the numerically largest cluster (cluster 2) shows values below this range.

## 4. Discussion

The principles of an analogue or virtual articulator, i.e., the spatial assembly of the maxilla, mandible and hinge axis to perform movements of the mandible in relation to the maxilla, remains an indispensable tool in dentistry. Regardless of all the controversies about the existence of a pure rotational movement [22] within the boundary movements of the mandible, the hinge axis continues to be an integral part of dental treatment concepts.

In orthognathic surgery, the situation is different from non-surgical oral rehabilitations that require planning in a (virtual) articulator, because the hinge axis determined preoperatively has a different importance than that determined in the postoperative situation. When planning an orthognathic procedure, the rotation of the mandible around the hinge axis is not only necessary for the fabrication of a surgical splint, but also when working with the concept of autorotation [23]. In a few cases, there is also an overlapping of the jaws during planning, which can be solved by rotation [22].

Although an accurate determination of the maxillo-mandibular relation is mandatory in orthognathic surgery planning, this relation is lost during the surgical procedure. By separating the mandible into proximal joint-bearing segments and a distal tooth-bearing segment, maintaining a physiological joint position seems to be complex. The condyles adopt a new position during surgery [24], and the postoperative shape of the mandible results in altered function of the adjacent musculature. Condylar positioning devices fail to maintain the preoperative situation, and manual repositioning of the proximal segment continues to be the method of choice [25]. This non-obvious inaccuracy justifies, in our understanding, the use of arbitrary mounting values if the clinical error size remains below 2 mm in translation and 4 degrees in rotation [26,27].

Extracting the hinge axis and its corresponding distance to the upper jaw from the lateral cephalogram is a separate process that contributes to the overall planning time. To save this step and to facilitate the mounting of intraoral scans in virtual space, a search for corresponding patterns in patients who had undergone orthognathic surgery was conducted. We could identify three patient groups with significantly different mounting values, which also differ in terms of the skeletal parameters (Figure 4).

Based on the results, we concluded that the virtual maxilla of vertical face class II (predominantly female) patients can be arbitrarily mounted at an angle of (rounded) $11^{\circ }$ to the AOP, with the hinge axis 27 mm vertical (AxV) and 88 mm horizontal (AxH) to the upper incisor edge. Class III horizontal face patients can be mounted at an angle of $2^{\circ }$ to the AOP, with the hinge axis 36 mm vertical and 86 mm horizontal to the upper incisor edge. All other (predominantly male) patients, with no clear tendencies to class II, class III, horizontal or vertical face cases, can be mounted at an angle of $8^{\circ }$ to the AOP, with the hinge axis 36 mm vertical and 99 mm horizontal to the upper incisor edge (Table 6). The application of the presented mounting method is shown in Figure 5.

The results of this study show that the Bonwill and Balkwill averages are not suitable to arbitrarily mount the jaws of a group of patients in need of orthognathic surgery. It can be assumed that these patients differ from the average study participants of Bonwill and Balkwill due to their skeletal malocclusion. A study regarding Bonwill and Balkwill averages in 120 randomly selected CBCT datasets of patients who did not undergo orthognathic surgery supports our findings [28]. The authors found similar values to Bonwill and Balkwill, namely a symmetrically mean arm of 103.3 mm with a smaller base of 99.6 mm between the condyles and a mean angle between the occlusal plane and the Bonwill arms of 20.4°. In our study, these values correspond to those of cluster 1, which is characterized by a balanced face (Table 5).

Articulators have been used for decades in orthognathic surgery planning with excellent clinical success [11,29]. The measured error rates of conventional plaster planning are within one millimeter, on average [30]. Moreover, the stability and predictability of orthognathic surgery depends not only on planning, but also on the direction of surgical movement, type of fixation and the surgical technique used [26]. These arguments support the use of the hinge axis for digital planning, especially when using surgical splints, whose clinical accuracy has not yet been surpassed [31].

The decision to use a planning system based on a physical articulator, a virtual articulator or a CBCT is not a decision relating to accuracy, as all systems provide similar results [11] and remain below the general accepted clinical error size of two millimeters in translation [26] and four degrees in rotation [27].

The present study has several limitations. First of all, it is important to mention the retrospective nature of the study and the specific patient cohort, which may vary from clinic to clinic. The values determined in this study can therefore only be transferred to other cohorts to a limited extent. The evaluated records are taken from routine patient care, so that an inter-individual magnitude of error must be assumed in the analysis of the radiographs, both for cephalometry and for virtual mounting. However, a strength of the study can be seen in the high number of included patients. This ensures that different skeletal malocclusions are represented in large numbers.

## 5. Conclusions

This is the first study to provide arbitrary values to virtually mount intraoral scans for orthognathic surgery planning. The obtained data on the position of the hinge axis can be applied to any digital planning in orthognathic surgery using CBCT or a virtual articulator, provided that the case to be planned can be clearly assigned to one of the clusters. Bonwill and Balkwill averages are not suitable for orthognathic surgery planning.

## Figures and Tables

**Figure 1 jcm-12-03582-f001:**
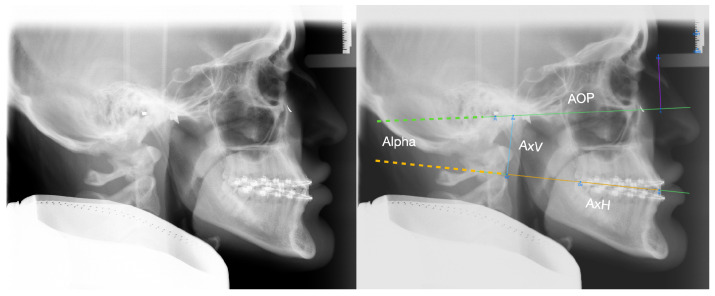
Left: lateral cephalogram with metal markers indicating the clinical position of the facebow. Right: calculation of mounting data based on the landmarks Pi, Ns, Ie, Dc and the dimensions of the SAM (SAM Präzisionstechnik GmbH, Gauting, Germany) facebow. The landmarks Ref0 and Ref1 are used for the scaling of the cephalogram. The plane of the facebow indicates the axis orbital plane (AOP; green line) represented by the line Pi-b, where b is the intersection point on the AOP perpendicular to Ns. One angle and two distances were constructed: the angle α between the upper occlusal plane (Ie-Dc; dotted orange line) and the AOP (Pi-b), the vertical distance (AxV; blue line) between the arbitrary hinge axis (Ax) and its projection (Ax’) onto the occlusal plane, and the horizontal distance (AxH; orange line) between Ie and Ax’.

**Figure 2 jcm-12-03582-f002:**
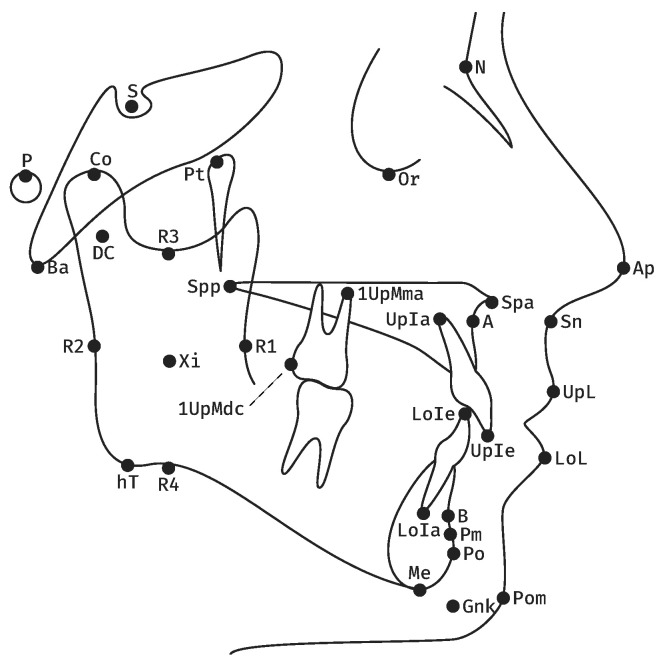
Landmarks used in the 22-item cephalometric analysis of the University of Münster: Nasion (N), Basion (Ba), Orbitale (Or), Porion (P), Pterygoid point (Pt), Sella (S), Anterior nasal spine (Spa), Posterior nasal spine (Spp), A point (A), Condylion (Co), Condylar midpoint (DC), Anterior border of the Ramus (R1), Posterior border of the Ramus (R2), Semilunar incisure (R3), Lower border of the Ramus (R4), Ramus midpoint (Xi), Menton (Me), Pogonion (Po), B Point (B), Suprapogonion (Pm), Constructed gnathion (Gnk), Upper Incisor edge (UpIe), Upper Incisor apex (UpIa), Lower Incisor edge (LoIe), Lower Incisor apex (LoIa), First Upper Molar mesial apex (1UpMma), First Upper Molar distal contact (1UpMdc), Apex nasi (Ap), Subnasal (Sn), Upper Lip (UpL), Lower Lip (LoL) and Pogonion molle (Pom). The angles and distances of the analysis are described in Table 1.

**Figure 3 jcm-12-03582-f003:**
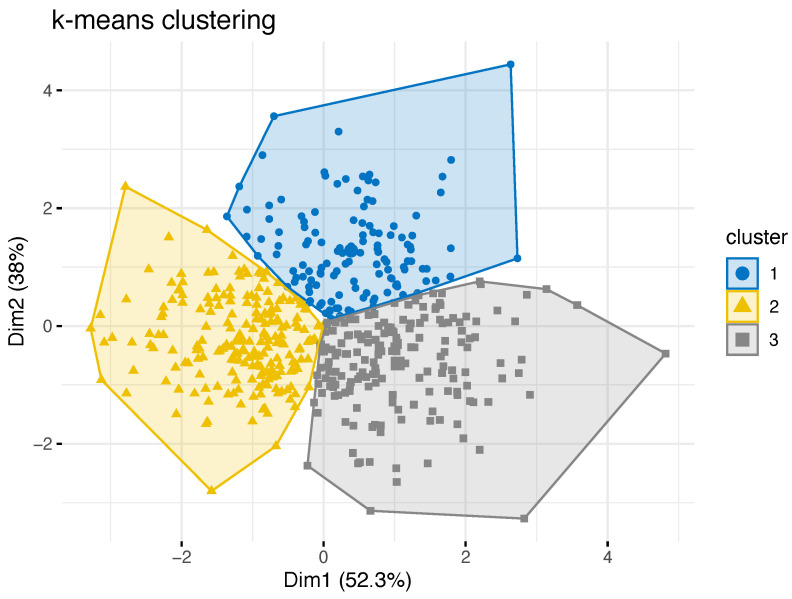
Cluster visualization. Dim1 and Dim2 represent the first two principal components of the three mounting parameters. The largest cluster is cluster 2, with 204 patients, followed by cluster 3 with 182 patients and cluster 1 with 128 patients.

**Figure 4 jcm-12-03582-f004:**
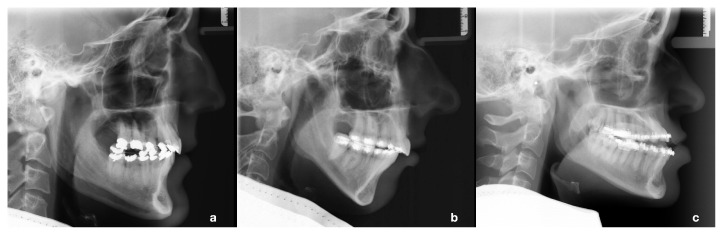
Example of cluster-specific skeletal phenotypes. (**a**) Cluster 1. Male patient with balanced jaw relationship. Surgical procedure was 2-piece maxilla with posterior impaction and slight maxillo-mandibular advancement. (**b**) Cluster 2. Female patient with vertical face and class II. Surgical procedure was maxillary impaction and mandibular advancement. (**c**) Cluster 3. Female patient with horizontal face and class III. Surgical procedure was maxillary advancement with posterior impaction and asymmetric mandibular set back.

**Figure 5 jcm-12-03582-f005:**
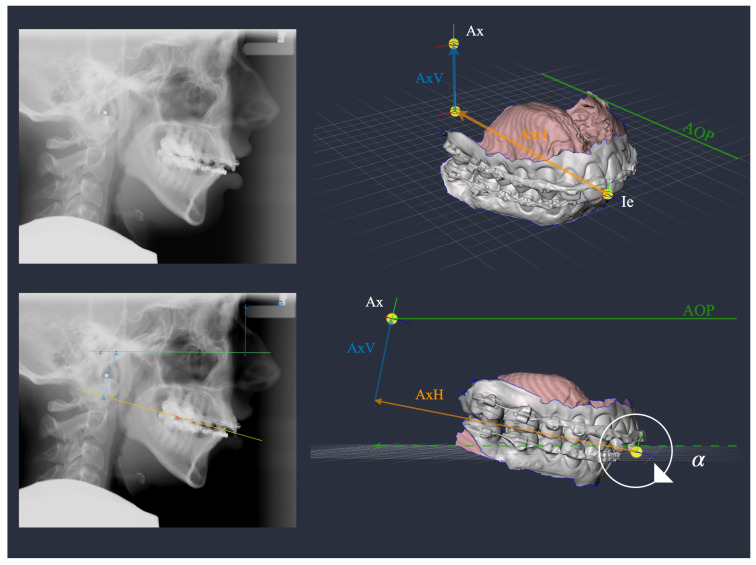
Application of arbitrary mounting data on a vertical face skeletal class II patient, which could be attributed to cluster 2. Upper right: the scans are oriented anatomically to the ground plane grid, which is parallel to the axis orbital plane (AOP; green line) and perpendicular to the mid-sagittal plane. The upper occlusal plane (uOP) must be consistent with the ground plane. The landmark upper incisor edge (Ie) is placed in the mid-sagittal plane in the level of the grid. Ie is then shifted posteriorly along the midline by 88 mm (AxH; orange line), and afterwards, shifted superiorly by 27 mm (AxV; blue line). In doing so, the position of the hinge axis (Ax) is determined. The final step is the rotation of the whole scene by the degree of α=11°. The center of rotation lies in Ie on the ground plane. Now, the models are correctly oriented according to a virtual facebow.

**Table 1 jcm-12-03582-t001:** Cephalometric angles and distances to evaluate inter-cluster differences. This analysis consists of 22 items and is the cephalometric analysis developed at the University of Münster. It is a combination of cephalometric analyses from Ricketts [18], Rakosi [19], Downs [20] and Steiner [21]. Landmarks are provided in Figure 2.

Measurement		Definition
1	Facial axis	Posterior angle btw. Ba-N and Pt-GnK.
2	Facial depth	Posterior angle btw. P-Or and N-Po.
3	SNB	Posterior lower angle btw. S-N and N-B.
4	Mandibular plane	Anterior angle btw. P-Or and hT-Me.
5	Inner gonion angle	Anterior angle btw. DC-Xi and Xi-Pm.
6	Relative mandibular length	Length of Co-Po.
7	Maxillary position	Posterior lower angle btw. Ba-N and N-A.
8	SNA	Posterior lower angle btw. S-N and N-A.
9	Palatal plane	Anterior angle btw. P-Or and Spa-Spp.
10	Rel. max. length	Length of Co-A.
11	Lower facial height	Anterior angle btw. Spa-Xi and Xi-Pm.
12	Convexity of point A	Distance btw. A and N-Po.
13	Rel. max. to mand. length	Ratio btw. Co-A and Co-Po.
14	Lower Incisor position	Distance btw. LoIe and A-Po.
15	Lower Incisor inclination	Caudal angle btw. LoIe-LoIa and A-Po.
16	Upper Incisor position	Distance btw. UpIe and A-Po.
17	Upper Incisor inclination	Caudal angle btw. UpIe-UpIa and A-Po.
18	Inter-Incisor angle	Anterior angle btw. UpIe-UpIa and LoIe-LoIa.
19	Vertical molar distance	Distance btw. 1UpMma and Spa-Spp.
20	Sagittal molar distance	Distance btw. 1UpMdc and a vertical to P-Or from Pt.
21	Lower Lip to E-Line	Distance btw. LoL and Ap-Pom.
22	Upper Lip Drape	Posterior angle btw. UpL-Sn and P-Or.

**Table 2 jcm-12-03582-t002:** Baseline characteristics of the study group divided by gender: age (in years), angulation (α) between the upper occlusal plane and the axis orbital plane (in degrees), vertical distance (AxV) between the arbitrary hinge axis (Ax) and its projection (Ax’) onto the occlusal plane, and the horizontal distance (AxH) between the upper incisor edge and Ax’ (in mm).

	Female			Male			*p*
	n	M	SD	n	M	SD	
age	323	26.1	8.7	191	27.9	8.3	<0.001
α	323	7.6	5.3	191	6.5	5.0	0.011
AxV	323	30.2	5.7	191	35.8	5.7	<0.001
AxH	323	88.0	6.7	191	93.1	8.1	<0.001

M = mean value; SD = standard deviation.

**Table 3 jcm-12-03582-t003:** Age (in years), angulation (α) between the upper occlusal plane and the axis orbital plane (in degrees), vertical distance (AxV) between the arbitrary hinge axis (Ax) and its projection (Ax’) onto the occlusal plane, and the horizontal distance (AxH) between the upper incisor edge and Ax’ (in mm) according to the three clusters.

		Cluster 1			Cluster 2			Cluster 3			*p*
		n	M	SD	n	M	SD	n	M	SD	
age		128	28.1	9.1	204	26.2	7.8	182	26.5	9.0	n. s.
	female	36	26.3	9.2	171	26.0	7.9	116	26.3	9.7	n. s.
	male	92	28.8	9.0	33	27.6	7.0	66	26.9	7.9	n. s.
α		128	7.8	3.4	204	11.1	3.3	182	2.4	4.0	<0.001
AxV		128	36.2	4.6	204	26.5	3.8	182	35.9	4.5	<0.001 ^1^
AxH		128	98.9	5.5	204	88.1	5.4	182	85.6	5.6	<0.001

^1^ Not significant between cluster 1 and 3. M = mean value; SD = standard deviation.

**Table 4 jcm-12-03582-t004:** Nine remaining cephalometric variables (as defined in Table 1) that differ statistically significantly in each cluster. Thirteen variables were excluded because they differed in only two of three clusters.

	Cluster 1		Cluster 2		Cluster 3		*p*
	*n* = 128		*n* = 204		*n* = 182		
	M	SD	M	SD	M	SD	
Facial axis	89.1	5.7	85.7	5.9	91.8	6.5	<0.001
Facial depth	89.5	4.8	85.7	5.0	92.1	5.3	<0.001
SNB	80.2	5.6	76.0	5.2	82.4	6.4	0.002 ^a^
Rel. mand. length	118.7	10.0	106.3	9.0	114.0	9.2	<0.001
SNA	82.4	4.3	79.9	4.3	81.1	4.9	0.043 ^a^
Convexity of point A	1.3	4.8	2.8	5.0	−2.8	4.6	0.012 ^b^
Rel. max. to mand. length	1.4	0.1	1.3	0.1	1.4	0.1	0.002 ^a^
Sagittal molar distance	20.4	4.8	16.9	4.2	19.1	4.5	0.034 ^a^
Lower Lip to E-Line	−2.2	3.6	−1.3	3.0	−3.3	3.3	0.036 ^b^

^a^ Lowest significance between clusters 1 and 3. ^b^ Lowest significance between clusters 1 and 2. M = mean value; SD = standard deviation.

**Table 5 jcm-12-03582-t005:** The distance of the upper incisor edge (Ie) to the hinge axis (Ax) as an approximation of the Bonwill arm (in mm; norm value 101.6 mm) as well as the Balkwill angle (in degrees; norm range 18°–25°) according to the three clusters.

	Cluster 1		Cluster 2		Cluster 3		*p*
	*n* = 128		*n* = 204		*n* = 182		
	M	SD	M	SD	M	SD	
Ie-Ax	105.4	5.7	92.1	5.3	92.9	5.9	<0.001 ^1^
Balkwill	20.1	2.4	16.8	2.4	22.8	2.5	<0.001

^1^ Not significantly different between clusters 2 and 3. M = mean value; SD = standard deviation.

**Table 6 jcm-12-03582-t006:** Mounting values by skeletal phenotypes. Angulation (α) of the upper occlusal plane to the axis orbital plane (in degrees), vertical distance of the hinge axis to the occlusal plane (AxV), and horizontal distance (AxH) from AxV to the upper incisor edge (in mm).

Cluster	Skeletal Phenotype	α	AxV	AxH
1	balanced face	8	36	99
2	vertical face, class II	11	27	88
3	horizontal face, class III	2	36	86

## Data Availability

The data presented in this study are available on reasonable request from the corresponding author. The data are not publicly available due to privacy reasons.

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
