# Peer review of "The Position of the Virtual Hinge Axis in Relation to the Maxilla in Digital Orthognathic Surgery Planning—A k-Means Cluster Analysis"

_jcm, 2023, doi:10.3390/jcm12103582_

Round 1

Reviewer 1 Report

Re: jcm-2376383

The position of the virtual hinge axis in relation to the maxilla in orthognathic surgery patients- a k-means cluster analysis

This study concluded that Balkwill angle and Bonwill triangle are not useful for orthognathic surgical planning. It was difficult to understand how this conclusion was induced from the result of this study. The materials and methods section and results section were confused and unclear. The k-mean cluster analysis should be main point based on the title. However, there was little explanation about the analysis. There was not detail about the methods, who and how examined, and what was examined on the lateral cephalogram. The number of table 5 and 6 may be opposite. Totally, the text should be reconstructed easier for readers.

Author Response

Dear Reviewers,
Thank you for taking the time to review our manuscript “The Position of the Virtual Hinge Axis in Relation to the Maxilla in Orthognathic Surgery Patients - A K-means Cluster Analysis”. We are grateful for your feedback and value your constructive suggestions. We hope that your concerns and suggestions were addressed appropriately. In the following, we give a point-by-point reply to your comments. You will find changes throughout the body of the manuscript. Thank you once again for your support to improve our manuscript. We hope
that you recommend our paper for publication.

On behalf of all co-authors
Sincerely,
Jonas Q. Schmid

Reviewer 1

Point 1: This study concluded that Balkwill angle and Bonwill triangle are not useful for orthognathic surgical planning. It was difficult to understand how this conclusion was induced from the result of this study.

Response: Dear reviewer, thank you for your insightful comment. We agree with you that the derivation of this conclusion could be explained in more detail. Therefore, we have extended the paragraphs about the Bonwill and Balkwill averages in the Introduction (lines 22-28, 76-81), Material and Methods (lines 110-113), results (lines 181-186), and Discussion (lines 226-235). To increase the efficiency of orthognathic surgery planning, the question arose whether Bonwill and Balkwill averages are sufficient to define the position of the hinge
axis. The results of this study show that the Bonwill arm and Balkwill angle in a cohort of patients who underwent orthognathic surgery are quite different from the average values propagated by Bonwill and Balkwill (Table 5). In our opinion, this difference justifies the conclusion that Bonwill and Balkwill averages are not suitable for mounting models in orthognathic surgery planning.

Point 2: The materials and methods section and results section were confused and unclear. The k-mean cluster analysis should be main point based on the title. However, there was little explanation about the analysis. There was not detail about the methods, who and how examined, and what was examined on the lateral cephalogram.

Response: Thank you very much for your feedback. We have done our best to make the manuscript easier to understand. Therefore, we have rewritten large parts of the Material and Methods and Results sections. We agree with you that the cluster analysis needs to be an integral part of the manuscript. To facilitate reproducibility and to increase comprehensibility, we have added step-by-step instructions on how the k-means cluster analysis was performed (lines 115-124). Furthermore, we referred more precisely to the cephalometric values in Table
1, which were used to identify cluster specific skeletal phenotypes. We agree with you that we could be more precise about what was measured on the cephalogram. For this reason, we have specified the landmarks of the cephalometric analysis in more detail in figure 2 and expanded the heading of table 1. 

Point 3: The number of table 5 and 6 may be opposite.
Response: Thank you for bringing this inconsistency to our attention. We have changed the order of table 5 and 6 and want to apologize for this avoidable mistake. 

Point 4: Totally, the text should be reconstructed easier for readers.
Response: Dear reviewer, thank you very much for your honest words that help us to improve our manuscript. This is undoubtably a complex topic, but we have tried our best to make the manuscript easier to understand. To do so, we have rewritten large parts of the Introduction, Material and Methods, Results, and Discussion. We hope that we have been able to improve the line of argumentation throughout the manuscript. We have put particular emphasis on
bringing the key messages in the manuscript more into focus. The great achievement of our study is that we can provide average mounting values for planning orthognathic surgery that depend on the skeletal configuration of the patient. The obtained data on the position of the hinge axis can be applied to any digital planning in orthognathic surgery using CBCT or a virtual articulator. 

Reviewer 2 Report

Please use a word document and write the answers in blue in the follow manner:

Comment (reviewer):

Response:

Revised text:

Title :

Could be written better in order to highlight the aim of this study.

Abstract:

Overall impression: The article could be really interesting if it was better written. Reading the abstract the article seems interesting but going through the manuscript some shortcomings are observed. Please review the abstract after improving the materials and methods section.

Actually my suggestion in the text is about the follow sentence: “Balkwill angle and Bonwill triangle are not suitable for planning orthognatic surgery”. In order to understand better reading, I advice to delete it because the meaning should be contextualized in the discussion.

Introduction:

Overall impression: This section should be re-evaluated in order to be more educational for the reader.

Please delete this sentence :”In orthognathic surgery the situation is far more comfortable than it appears at first glance.” because not appropriate in scientific language.

The follow sentence is not appropriate because is different from the purpose stated in the abstract:

“The aim of the method was to achieve a high level of precision in surgical planning without exceeding the budget and time frame.”

Material and Methods:

Too short description. The inclusion and exclusion criteria are unclear and are not reported in table II (table II seems to refer to the measurements)

It is unclear how the authors conducted the study. Please describe the procedure in detail. It seems that a software was used, but it was not mentioned which software and for what purpose?

Discussion

“The concept of the “articulator”. Please rephrase and expand better. The articulator is not a concept. What do the authors mean?

The discussion section can be evaluated after a major revision of material and methods, for this reason it’s not judgeable.

Please improve english language using a scientific approach

Author Response

Dear Reviewers,
Thank you for taking the time to review our manuscript “The Position of the Virtual Hinge Axis in Relation to the Maxilla in Orthognathic Surgery Patients - A K-means Cluster Analysis”. We are grateful for your feedback and value your constructive suggestions. We hope that your concerns and suggestions were addressed appropriately. In the following, we give a point-by-point reply to your comments. You will find changes throughout the body of the manuscript. Thank you once again for your support to improve our manuscript. We hope that you recommend our paper for publication. 

On behalf of all co-authors
Sincerely,
Jonas Q. Schmid

Reviewer 2

Point 1: Title: Could be written better in order to highlight the aim of this study.
Response: Dear reviewer, thank you for your comment that helps to improve the manuscript. We have changed the title of the study to: “The Position of the Virtual Hinge Axis in Relation to the Maxilla in Digital Orthognathic Surgery Planning - A K-means Cluster Analysis“. This was done to emphasize one of the main messages of the paper, which is that the obtained mounting values can be used in the future by many clinicians and researchers involved in orthognathic surgery.

Point 2: Abstract: Overall impression: The article could be really interesting if it was better written. Reading the abstract the article seems interesting but going through the manuscript some shortcomings are observed. Please review the abstract after improving the materials and methods section.
Response: Thank you for your helpful comments and for recognizing the potential of our article. We appreciate that you find the abstract interesting. To eliminate the mentioned shortcomings, a great effort was made to extensively rewrite the article and make it more understandable. We have rewritten large parts of the Introduction, Material and Methods, Results, and Discussion. Regarding the Material and Methods section, we have added stepby-step instructions on how the k-means cluster analysis was performed (lines 115-124). The main achievement of our study is the providing of average mounting values for planning orthognathic surgery that depend on the skeletal configuration of the patient. These values on the position of the hinge axis can be applied to any digital planning in orthognathic surgery using CBCT or a virtual articulator. This finding was emphasized in the abstract (lines 11-13).

Point 3: Actually my suggestion in the text is about the follow sentence: "Balkwill angle and Bonwill triangle are not suitable for planning orthognatic surgery". In order to understand better reading, I advice to delete it because the meaning should be contextualized in the discussion.
Response: Dear reviewer, thank you for this important advice. We agree with you that this sentence is out of context. Due to the limitation of 200 words in the abstract, we were not able to describe the comparison with the Bonwill and Balkwill averages with enough detail. And have therefore removed this sentence from the abstract. As recommended, we have extended the paragraphs about the Bonwill and Balkwill averages in the Introduction (lines 22-28, 76-81), Material and Methods (lines 110-113), results (lines 181-186), and Discussion
(lines 226-235). The results of our study show that the Bonwill and Balkwill values in a cohort of patients who underwent orthognathic surgery are significantly different from the average values found by Bonwill and Balkwill, as further outlined in the discussion (lines 226-235).

Point 4: Introduction: Overall impression: This section should be re-evaluated in order to be more educational for the reader.
Response: Thank you very much for your feedback. We have tried our best to make the introduction more educational for the reader. To do so, we have rewritten large parts of it and hope that we have been able to improve the explanations in this section. We have placed particular emphasis on the definition of arbitrary mounting of models, the importance of the hinge axis, and the origin of the Bonwill and Balkwill averages. Furthermore, we have clarified the aim of the study.

Point 5: Please delete this sentence "In orthognathic surgery the situation is far more comfortable than it appears at first glance." because not appropriate in scientific language.
Response: We agree with you that this wording does not meet scientific standards. As suggested, we have therefore deleted this sentence and added an explicit explanation to the discussion (lines 201-210).

Point 6: The follow sentence is not appropriate because is different from the purpose stated in the abstract: "The aim of the method was to achieve a high level of precision in surgical planning without exceeding the budget and time frame.” 
Response: Thank you for this important comment. We agree with the reviewer that this sentence may be misleading. It was our intention to explain the objective of the development of the DMMS system, not to describe the aim of the present study. We have deleted this sentence accordingly (lines 54-55).

Point 7: Material and Methods: Too short description. The inclusion and exclusion criteria are unclear and are not reported in table II (table II seems to refer to the measurements).
Response: Thank you for your valuable feedback. To accommodate your recommendation, we have sharpened the inclusion and exclusion criteria (lines 100-104). We have also improved the description of the k-means cluster analysis and how the values of table 2 were used for inter-cluster comparisons (lines 115-117). 

Point 8: It is unclear how the authors conducted the study. Please describe the procedure in detail. It seems that a software was used, but it was not mentioned which software and for what purpose?
Response: Dear reviewer, we are very grateful that you have brought this issue to our attention. This will allow us to significantly increase the quality of our paper. To facilitate reproducibility, we have further expanded the materials and method section by describing the method in detail (lines 105-112). We have also added a reference to the software used for the construction of the mounting values (lines 74-76).

Point 9: Discussion. "The concept of the "articulator". Please rephrase and expand better. The articulator is not a concept. What do the authors mean?
Response: Many thanks for this important note. Our intention was to explain the concept of the hinge axis and its importance in mandibular opening rotation. As suggested, we have rephrased this term and added an extensive explanation on the importance of the hinge axis (lines 189-199). We hope that this will provide greater clarity. 

Point 10: The discussion section can be evaluated after a major revision of material and methods, for this reason it's not judgeable.
Response: Dear reviewer, as previously stated, we have rewritten large parts of the Material and Methods section. We hope that our revision has significantly improved the comprehensibility of this part, so that the Discussion can be judged accordingly.

Point 11: Please improve english language using a scientific approach
Response: Thank you for your honest feedback. The manuscript received extensive English editing. We hope that the English language does now meet academic standards.

Round 2

Reviewer 1 Report

Now my points were completely revised. This article reaches the level of the journal.